# Fumonisin B_1_ Exposure Causes Intestinal Tissue Damage by Triggering Oxidative Stress Pathways and Inducing Associated CYP Isoenzymes

**DOI:** 10.3390/toxins17050239

**Published:** 2025-05-12

**Authors:** Changyu Cao, Weiping Hua, Runxi Xian, Yang Liu

**Affiliations:** 1College of Animal Science and Technology, Foshan University, Foshan 528225, China; h17825598460@163.com (W.H.); x13434805906@163.com (R.X.); 2Foshan University Veterinary Teaching Hospital, Foshan 528231, China; 3Quality Control Technical Center (Foshan) of National Famous and Special Agricultural Products (CAQS—GAP—KZZX043)/South China Food Safety Research Center, Foshan 528231, China; liuyang@fosu.edu.cn

**Keywords:** Fumonisin B_1_, intestinal, oxidative stress, Cytochrome P450 proteins, NXRs

## Abstract

Fumonisin B_1_ (FB_1_) is considered the most toxic fumonisin produced by fungi and is commonly found in contaminated feed and crops. Fumonisin and its metabolites extensively exist in feed and crops, where FB_1_-polluted crop ingestion can do harm to livestock and poultry, causing poultry intestinal toxicity in the latter. For investigating FB_1_-mediated intestinal toxicity, we assessed the function of FB_1_ exposure in quail intestines and explored its possible molecular mechanisms. In total, 120 quail pups were classified into two groups, where those in the control group were given a typical control diet, and those in the experimental group were given a typical diet that contained 30 mg/kg FB_1_. We evaluated the histopathological and ultrastructural changes in quails’ intestines on days 14, 28, and 42, and studied the molecular mechanisms by assessing oxidative stress, inflammation, and nuclear xenobiotic receptors (NXRs). Our results suggest that FB_1_ exposure causes intestinal inflammation by triggering oxidative stress pathways and modulating NXRs to induce Cytochrome P450 proteins (CYP) isoforms, leading to intestinal histopathological damage. The results of this study shed novel light on the molecular mechanism underlying FB_1_-induced intestinal injury in juvenile quails.

## 1. Introduction

Fumonisins are produced by several *Fusarium* species, such as *F. proliferatum*, and are fatal mycotoxins that contaminate feed and cause various animal diseases. Typically, B-type fumonisins are considered the most toxic [1]. They can be toxic to the liver, kidney, and embryo, and induce arterial plaque formation and immunosuppression in experimental animal systems (mice and rats) [2] (pp. 9481−9515). High levels of exposure to Fumonisin B_1_ (FB_1_) are toxic to the liver and also cause gastrointestinal injury in mice and rats [3] (pp. 185−206). Although its toxic mechanism is extensively investigated among experimental animals, research on birds is lacking, and the specific effect of FB_1_ on quail intestines is unclear.

The intestinal barrier plays a crucial role in preventing invasion by enteric pathogens, commensal bacteria, or natural toxins [4,5,6]. When food contaminated with FB_1_ is consumed, the intestines are the first organ to be exposed and damaged. Once the intestinal barrier is compromised, toxins present in the intestine can enter the bloodstream, leading to toxin translocation and promoting intestinal infections. Moreover, these toxins can directly affect other organs and may even trigger a systemic inflammatory response syndrome [7].

Cellular mechanisms underlying FB_1_-mediated toxicity are related to inducing cytotoxicity, oxidative stress, and changes in cytokine production in mice, chick, and intestinal antigen-presenting cells in pigs [1,8,9,10]. The functions of FB_1_ in intestines include interference with homeostasis of nuclear xenobiotic receptors (NXRs), causing inflammation and histopathology (Length of villi, depth of crypts, ratio of villi to crypts, number of goblet cells, neutrophils and lymphocytes; morphology of mitochondria, etc.), and have been long demonstrated in mice [11,12,13]. However, the phenomenon and mechanism of intestinal damage caused by FB_1_ in poultry are not clear. In this study, we used the toxicological model animal quail as the research object; the effects of FB_1_ on the intestines of quails were studied from the aspects of oxidative stress, inflammation, and NXRs.

## 2. Results

### 2.1. Histopathological Observation for Ileum Intestinal Tissue

Pathological sections of quail ileum with HE staining are shown in Figure 1. Intestinal villus length was not significantly different in experimental versus control groups at 14 days. At 28 days, villus length increased in the experimental group relative to the control group (*p* < 0.05). At 42 d, the intestinal villus length of the experimental group remarkably decreased relative to the control group (*p* < 0.05) due to partial intestinal villous shedding and necrosis. FB_1_ exposure resulted in an increase in intestinal crypt depth compared to the control group (*p* < 0.05), leading to a decrease in the V/C ratio in the FB_1_ group relative to the control group (*p* < 0.05). Goblet cells, neutrophils, and lymphocytes quantities of the experimental group increased relative to the control group (*p* < 0.05).

### 2.2. TEM Ultrastructural and Fibrous Changes in the Ileum

Relative to the control group, microvilli at the top of the basement membrane epithelial cells in the FB_1_ group sloughed off, with the effect increasing with an increase in days of exposure to FB_1_ (Figure 2B,D,E). The FB_1_ group also showed different degrees and numbers of mitochondrial swelling and inner ridge shortening (Figure 2B,D,E). This suggests that FB_1_ exposure induced ileum injury.

### 2.3. Functions of FB_1_ in Oxidative Stress-Associated mRNA Expression

Figure 3 showed that relative to the control group, the Nrf2 and NQO-1 mRNA expression of the 14d group apparently decreased (*p* > 0.05). The HO-1 and NQO-1 mRNA expression of the 28d group also apparently decreased; likewise, the Nrf2, HO-1, and NQO-1 mRNA expression of the 42d group also decreased (*p* < 0.05). The HO-1 mRNA expression of the 14d group and Nrf2 mRNA expression of the 28d group did not significantly change (*p* < 0.05). This suggests that exposure to FB_1_ induces oxidative stress in the quail intestine.

### 2.4. Functions of FB_1_ in Oxidative Stress and Inflammatory Signaling Pathways

According to Figure 4, the TLR4, NF-kB, TNF-α, COX-2, and iNOS mRNA expression of experimental groups markedly increased relative to the control group (*p* < 0.05), indicating FB_1_-induced intestinal inflammation in quail.

### 2.5. Functions of FB_1_ in Heterologous Nuclear Receptors mRNA Expression and Related Subunits

As shown in Figure 5, the AHR mRNA expression of the experimental group markedly increased relative to the control group (*p* < 0.05). The CYP1A5 and CYP1B1 mRNA expression of the experimental group remarkably increased relative to the control group (*p* < 0.05), and the CYP1A1 mRNA level evidently decreased. CYP1A4 mRNA expression was not markedly changed in the 14d and 28d groups, but in the 42d group, expression apparently decreased (*p* < 0.05).

From Figure 6, CAR and PXR, as well as the CYP3A4 and CYP3A9 mRNA levels of the experimental group, were significantly up-regulated relative to the control group (*p* < 0.05). The CYP2C18 mRNA expression of the 14d group markedly increased relative to the control group (*p* < 0.05), but no significant change was seen in the 28d group, while it dramatically decreased in the 42d group (*p* < 0.05).

## 3. Discussion

Oxidative stress indicates the imbalance between free radical production and antioxidant synthesis [14] (pp. 585−607). Studies have demonstrated FB_1_ toxicity, oxidative stress, and injury to the vital organs of broilers [12]. Similarly, oxidative stress is associated with a dysfunction of the intestinal barrier and a variety of digestive tract disorders. Its negative impacts on intestinal morphology usually occur together with decreased tight junction protein levels [15]. Similar to our results, FB1 exposure-induced oxidative stress was also found in chickens and mice in experiments by Devriendt et al. [11] (2009) and Frisvad et al. (2018). [2] (pp. 9481–9515) In quail, some redox genes also showed stronger species specificity.

Drawing from existing literature, we deduced that FB_1_ exposure in quail likely reduces the expression of genes related to intestinal oxidative stress. Nrf2 is a transcription factor essential for activating antioxidant defense mechanisms and maintaining redox homeostasis. When downregulated, as observed in our study, antioxidant responses may be impaired, contributing to oxidative damage. Our findings indicate a significant downregulation of Nrf2 along with its target gene products HO-1 and NQO1, particularly in the 42-day exposure group. This implies that FB_1_ may perturb redox equilibrium, leading to intestinal oxidative stress by inhibiting the Nrf2 pathway, a mechanism consistent with prior research.

FB_1_ is known to impact the CYP450 enzyme system in animals, leading to various damages [16,17,18] (pp. 47−53; pp. 50−55; pp. 185−194; pp. 104−111). Our study aimed to clarify how FB_1_ affects the intestinal CYP450 system in quail. Nuclear receptors (NXRs), including AHR, PXR, and CAR, are known to regulate the CYP enzyme system, which is integral to several metabolic pathways [19]. The CYP450 enzymes, key players in phase-I metabolism, are categorized based on their roles in endogenous substance metabolism and biosynthesis, with CYP1-CYP4 families being activated by heterologous substances through NXR-dependent mechanisms. AHR, upon activation by ligands such as polycyclic aromatic hydrocarbons and certain clinical agents, enters the nucleus and binds to the Xenobiotic response element (XRE), inducing CYP1A1, CYP1A4, CYP1A5, and CYP1B1 expression [20] (pp. 38−54). PXR and CAR also regulate CYP3 and CYP2 subunits upon activation by their respective ligands [21] (pp. 326−333). Our examination of genes related to the CYP450 enzyme lineage, in conjunction with our prior research on NXRs, revealed that FB_1_ significantly affected CYP450 enzyme mRNA levels in quail intestine. This impact was associated with the activity of the NXRs AHR, CAR, and PXR, aligning with existing studies.

Inflammation in the intestine often involves Toll-like receptors (TLRs), which are targeted by regulatory mechanisms to control inflammation [22] (pp. 972-978). Activation of TLR4, a complex process, triggers the transcription of inflammatory genes and is regulated by pathways including JNK, MAPK, and NFĸB [23] (pp. 725−733). Our study of gene expression related to FB_1_ revealed a significant upregulation of TLR4 and inflammatory factors like NF-kB, TNF-, COX-2, and iNOS in experimental groups, with increased expression with longer exposure times. This indicates that FB_1_ may induce intestinal inflammation via the TLR4 pathway and other inflammatory mediators.

Given the intestinal inflammation caused by FB_1_ exposure in quail, we investigated the presence of pathological and ultramicroscopic damage in intestinal tissues using light and electron microscopy. The intestinal wall consists of four layers: mucosa, submucosa, muscularis, and serosa. The mucosa, critical for absorption, immunity, and barrier function, is divided into the epithelial and intrinsic layers and contains various cell types, including goblet, columnar, immune, endocrine, and stem cells, which form the intestinal glands. The muscularis, with its two smooth muscle layers, aids gland secretion. The villus length to gland depth ratio (V:C) indicates the small intestine’s digestive and absorptive capacity; a decrease suggests mucosal damage and reduced function due to FB_1_ exposure. The control group’s ileum showed a complete gland structure with orderly epithelial cells and well-developed villi. In contrast, the experimental group exhibited disordered epithelial cells, incomplete glands, and signs of necrosis and inflammation. The 28-day group displayed heightened villus hyperplasia and inflammation, while the 42-day group showed sparser villi with necrosis and inflammatory cell infiltration. These findings indicate that FB_1_ induces pathological damage in quail intestines, with severity increasing with exposure duration.

## 4. Conclusions

Collectively, this work demonstrates the functions of FB_1_ in the intestines of quails, including oxidative stress, the activation of NXRs and the CYP450 enzyme lineage, as well as inflammation and histopathological changes in Figure 7. Results showed that FB_1_ enters the quail and causes pathological damage to the intestinal tract, and the damage becomes more severe over time. These results suggest that FB1-induced enterotoxicity in quails involves the disruption of antioxidant defenses (via Nrf2 pathway inhibition), activation of nuclear receptors (AHR, PXR, CAR), and induction of intestinal inflammation (via TLR4-NFκB pathway), with severity increasing over time. Future research should focus on understanding the functions of FB_1_ in mucosal barrier function at a molecular level.

## 5. Materials and Methods

### 5.1. Animals and Treatments

The Animal Protection and Ethics Committee of Foshan University approved the experimental protocols. A total of 120 healthy, 1-day-old quail chicks were obtained from a commercial hatchery and raised under controlled conditions. Altogether 120 normal 1-day-old quails were obtained from hatching eggs bought from a hatchery farm and placed under the constant environment (original temperature at 36 ± 1 °C, later decreased as the days increased, and maintained at 29 ± 1 °C following 14 days; humidity 50 ± 15%; 12-h/12-h light/dark cycle). FB_1_ was purchased from Shanghai Yuanye Biotechnology Co., Ltd. (Shanghai, China) (purity ≥ 98%). The FB_1_ diet (30 mg/kg) was prepared by Guangdong Laboratory Animal Center (Feed information provided in the Appendix A). After seven days of acclimation, they were randomized into two groups, as follows: control, which received uncontaminated feed, and FB_1_, which received feed contaminated with 30 mg/kg FB_1_. On days 14, 28, and 42, quail were anesthetized, blood samples were obtained using a needle, and thereafter, the animals were humanely euthanized. After opening the abdominal cavity, the intestines were carefully dissected and cleaned with a pre-chilled saline solution, and then the small intestine was removed and placed in an EP tube, followed by immediate freezing with liquid nitrogen prior to subsequent analyses. Experimental datasets were classified as follows, 14d control group (C14, n = 20); Group 14d FB_1_ (F14, n = 20); 28d control group (C28, n = 20); 28d FB_1_ exposure group (F28, n = 20); Control group at 42 days (C42, n = 20); FB_1_ exposure group at 42 days (42 F, n = 20).

### 5.2. Light Microscope Examination

For microscopic observation, ileal tissue samples were subjected to fixation with 10% formalin solution, paraffin embedding, and cutting into 5 μm sections. Two sections were taken from each group of ileal tissues. After hematoxylin and eosin (H&E) staining, the pathological sections prepared were monitored with the microscope (China, mshot, ML31). The length V (villus height, V) of each villus and crypt depth C (Crypt depth, C) of each intestinal gland were measured digitally at 200× magnification, and the V:C ratio was calculated. In addition, the numbers of goblet cells, neutrophils, and lymphocytes were calculated from 3 visual fields at 400× magnification.

### 5.3. Transmission Electron Microscope Examination

Intestinal tissues (about 1 mm^3^ each) were quickly subjected to 3-h fixation using 2.5% glutaraldehyde contained within 0.1 M sodium phosphate buffer (pH 7.2), cleaning at 4 °C, then an additional 1-h fixation using 1% osmium tetroxide contained within sodium phosphate buffer. After dehydration for 10 min through 50% ethanol grading steps, with two changes of propylene oxide, the tissue was treated through aragonite embedding, then magnesium uranyl acetate and lead citrate were added to stain ultrathin sections, followed by observation by transmission electron microscopy (TEM, HITACHI HT7700 80 kv, USA).

### 5.4. Real-Time Fluorescence Quantitative PCR Assay

Through adopting TransZol Up Plus RNA Kit ^®^(TransGen Biotech, China), total RNA extraction (1) grinding tissue samples in liquid nitrogen; (2) adding TransZol Up Plus reagent; (3) centrifuging to separate RNA; (4) washing with 75% ethanol; (5) dissolving in water free of RNase. Total RNA was later subjected to resuspension in 30 μL RNase-free water. RNA content and quality were analyzed by spectrophotometry at 260/280 nm. Moreover, first-strand cDNA was synthesized based on total RNA with a reverse transcription kit^®^ preserved at −80 °C prior to subsequent analysis. Primers are shown in Table 1. The ABI 7500 system was used for qPCR analysis. The delta-delta Ct (2^−∆∆Ct^) approach was adopted to analyze the data (GraphPad Prism 8.0).

### 5.5. Statistical Analysis

The functions of different treatments in the ileal enterotoxicity rate among FB_1_-exposed quails were statistically analyzed by GraphPad Prism 8.0 and Microsoft Excel. Results were represented by mean ± standard deviation (S.D.). T-test and Tukey’s post hoc test were used for comparisons, with * *p* < 0.05 indicating statistical significance relative to the control group.

## Figures and Tables

**Figure 1 toxins-17-00239-f001:**
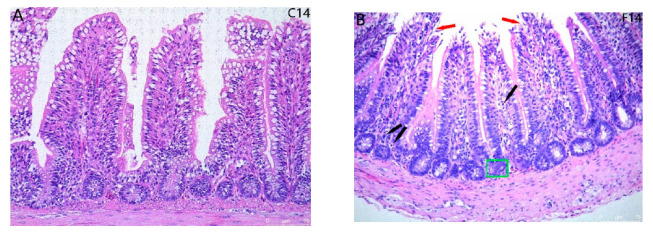
Pathological section observation results of quail ileum with HE staining. (**A**–**F**) HE staining pathological section (200×); Notes: red arrow indicates villous epithelial cell proliferation or villous shedding, black arrow indicates inflammatory cells, green box indicates necrotic intestinal gland; (**G**) Villus length V; (**H**) Intestinal gland depth C; (**I**) V:C; (**J**) The number of goblet cells; (**K**) Neutrophil count; (**L**) Number of lymphocytes. “*” indicates a significant difference between the experimental group and the control group, while “ns” indicates no statistical significance.

**Figure 2 toxins-17-00239-f002:**
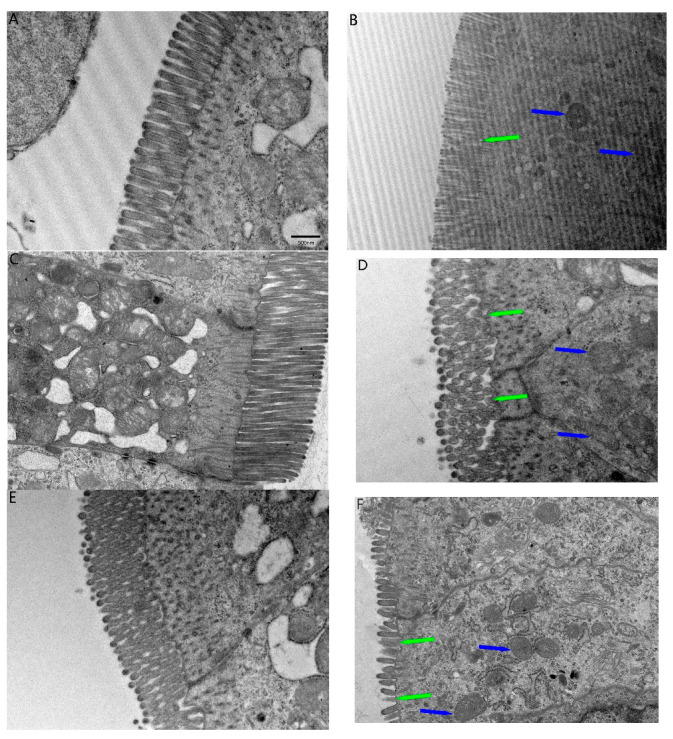
TEM observation of the ileum of quail exposed to FB_1_; Notes: Microvilli (green arrow) and mitochondria (blue arrow) were significantly negatively affected. (**A**) C14; (**B**) F14; (**C**) C28; (**D**) F28; (**E**) C42; (**F**) F42.

**Figure 3 toxins-17-00239-f003:**
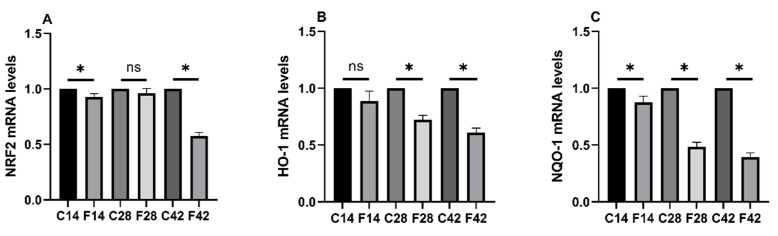
Ileum oxidative stress-related cytokine mRNA expression test results. (**A**) Nrf2; (**B**) HO-1; (**C**) NQO-1. Notes: “*” indicates a significant difference between the experimental group and the control group, while “ns” indicates no statistical significance.

**Figure 4 toxins-17-00239-f004:**
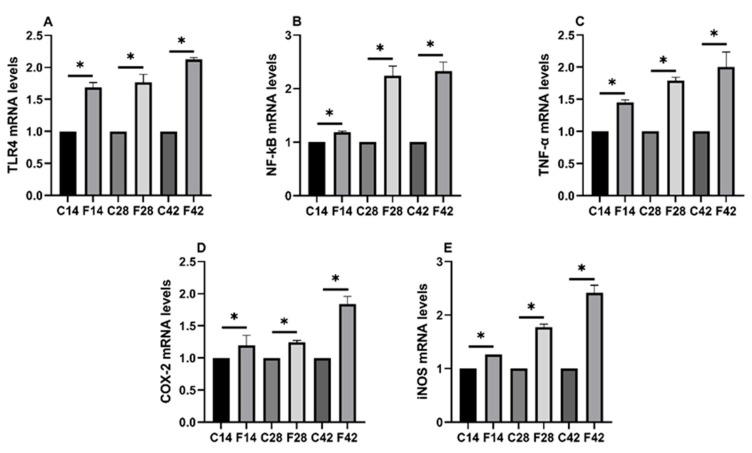
Inflammation-related cytokine mRNA expression levels in the ileum of quail exposed to FB_1_. (**A**) TLR4; (**B**) NF-kB; (**C**) TNF-α; (**D**) COX-2; (**E**) iNOS. Note: “*” indicates a significant difference between the experimental group and the control group.

**Figure 5 toxins-17-00239-f005:**
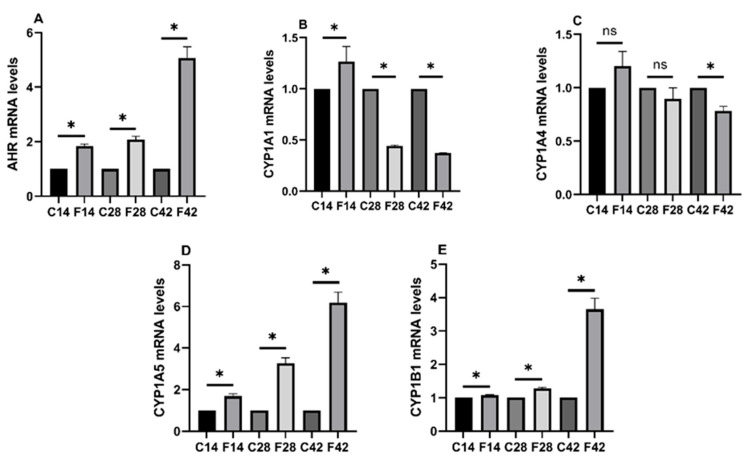
AHR and AHR-related CYP1 subunit mRNA expression test results. (**A**) AHR; (**B**) CYP1A1; (**C**) CYP1A4; (**D**) CYP1A5; (**E**) CYP1B1. Notes: “*” indicates a significant difference between the experimental group and the control group, while “ns” indicates no statistical significance.

**Figure 6 toxins-17-00239-f006:**
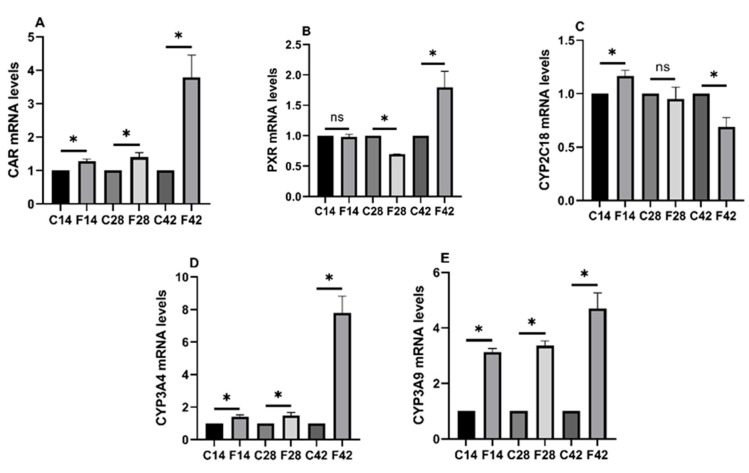
The mRNA expression assay results of CAR, PXR, and CAR, PXR-related CYP2 subunit, and CYP3 subunit within quail intestine after FB_1_ exposure. (**A**) CAR; (**B**) PXR; (**C**) CYP2C18; (**D**) CYP3A4; (**E**) CYP3A9. Notes: “*” indicates a significant difference between the experimental group and the control group, while “ns” indicates no statistical significance.

**Figure 7 toxins-17-00239-f007:**
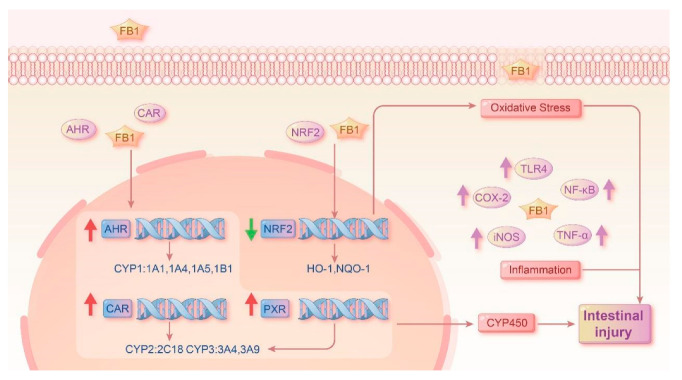
The pathway of FB_1_ in quail intestine and the resultant intestinal injury.

**Table 1 toxins-17-00239-t001:** Primer sequences in qRT-PCR.

Gene	Forward Primer (5′ to 3′)	Reverse Primer (5′ to 3′)
TLR4	CATCCCAACCCAACCACAGTAGCA	TGAGCAGCACCAACGAGTAGTATAGC
NF-κB	AGCAGAACTGAGAATTTGCC	CTGAACACTATTGCGACCTG
TNF-α	GTGTTCTATGACCGCCCAGT	TGTTCCACGTCTTTCAGAGC
COX-2	CCTATTACACAAGAAGCCTTCCACCAA	TCGCAGCAAGAATTTCTCCACAATCA
iNOS	TGTTATTAAGAACCAGCCCTC	CATCGGTATCTGCTTCTTGCT
Nrf2	CTCGCTCCAGTCCCGCTCGTA	GTGACTTCCCAGCCCTTGTCC
HO-1	ATGGAAACTTCGCAGCCACAC	CGTGACCAGCTTGAACTCGT
NQO-1	TATGAGATGGAGACGGCGCA	GAAAACGCGGTCAAACCAGC
JNK3	GCGAATGTCCTACCTGCTGTATCAA	CGAGTCACTACATAAGGCGTCATCAT
AHR	TTCAGGAAAGCAGAACAGCAA	TCACAACTAATACGAAGCCAT
CYP1A1	TTGCGTGTTTATCAACCAGT	CTTTGTTCACTTCGGTCCCTT
CYP1A4	ATGCTCGTTTCAGTGCCTTCGT	GTGTCAAAGCCTGCCCCAA
CYP1A5	CTATGACAAGAACAGCATCCGAGACT	CCCCAAAGATGTCATTCACC
CAR	ACTTCACCTGCCCCTTTGCC	CCTTCCTCATCCCCACGTCCA
PXR	CCCTCAAGAGCTACATCGACCA	TGTTCTCCATCTTCAGCGTCT
CYP2C18	AACCTCCATACGAAGCTGCAA	TGTGCCTTTGAAGACTTTCTCA
CYP3A4	TCATAGTGTTGTTCCCCTT	GGTATCCTTCTTCCCGTTC
CYP3A9	ATGCTCGTTTCAGTGCCTTCGT	GTGTCAAAGCCTGCCCCAA

## Data Availability

The original contributions presented in this study are included in the article/Appendix A. Further inquiries can be directed to the corresponding author.

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
