# Peer review of "Fumonisin B1 Exposure Causes Intestinal Tissue Damage by Triggering Oxidative Stress Pathways and Inducing Associated CYP Isoenzymes"

_toxins, 2025, doi:10.3390/toxins17050239_

Round 1
Reviewer 1 Report
Comments and Suggestions for Authors
Line 32: “The intestinal barrier plays an important role in invasion by invading enteric pathogens, commensal bacteria, or natural toxins (Wang et al., 2019). After eating the FB1-polluted food, the intestines are the initial organ to be exposed and damaged. When the intestinal barrier is damaged, the toxins in the intestine enter the blood, causing toxin displacement and promoting intestinal infection. Additionally, the toxins can now directly contact each organ and even cause systemic inflammatory response syndromes”, try to rephrase it, sound odd.
Line 55: “the oral normal feed (control) and the oral FB1 (including feed/FB1) groups.” Please find another phrasing maybe: 2 groups: Control, who received with uncontaminated feed, and FB1, who received feed contaminated with 30 mg/kg FB1. After opening the abdominal cavity, intestines were carefully dissected, and cleaned with a pre-chilled saline solution, and then the small intestine was removed and placed in an EP tube, followed by immediate freezing with liquid nitrogen prior to subsequent analyses. Again, odd phrasing.
In line 101 is an extra z in zincreased.
Line 104 Intestinal crypt depth of experimental group increased relative to control group (p<0.05 ) and thus V: C in experimental group decreased compared with control group (p < 0.05 ). Try something like: FB1 exposure led to an increase in crypt depth compared tocontrol group, or something to highlight that FB1 is responsible for observed changes.
3.4. Chapter TLR4 is not a cytokine, I’ve seen in the discussion part that you mentioned the receptors, please merge the oxidative and inflammatory response, or try whatever suits you, have you tried to see what is happening all over the signaling pathway? MD2, MYD88, IRAK, JNK, MAPK etc?
There is no conclusion chapter?
With a few improvements, the results obtained are very beautiful and will certainly have a positive impact for the journal.
Author Response
Q1:Line 32: “The intestinal barrier plays an important role in invasion by invading enteric pathogens, commensal bacteria, or natural toxins (Wang et al., 2019). After eating the FB1-polluted food, the intestines are the initial organ to be exposed and damaged. When the intestinal barrier is damaged, the toxins in the intestine enter the blood, causing toxin displacement and promoting intestinal infection. Additionally, the toxins can now directly contact each organ and even cause systemic inflammatory response syndromes", try to rephrase it, sound odd.
A1: Thank you for your valuable feedback and for the opportunity to refine our manuscript. I changed it to ‘The intestinal barrier plays a crucial role in preventing invasion by enteric pathogens, commensal bacteria, or natural toxins (Wang et al., 2019). When food contaminated with FB1 is consumed, the intestines are the first organ to be exposed and damaged. Once the intestinal barrier is compromised, toxins present in the intestine can enter the bloodstream, leading to toxin translocation and promoting intestinal infections. Moreover, these toxins can directly affect other organs and may even trigger a systemic inflammatory response syndrome (Dong et al., 2020).’(lines 43-49)
- Wang, W. C., Zhai, S. S., Xia, Y. Y., Wang, H., Ruan, D., Zhou, T., Zhu, Y. W., Zhang, H. F., Zhang, M. H., Ye, H., Ren, W. K., and Yang, L. (2019). Ochratoxin A induces liver inflammation: involvement of intestinal microbiota. Microbiome7.
- Dong, P., Zhang, Y., Yan, D. Y., Wang, Y., Xu, X., Zhao, Y. C., and Xiao, T. T. (2020). Protective Effects of Human Milk-Derived Exosomes on Intestinal Stem Cells Damaged by Oxidative Stress. Cell Transplantation29.
Q2:Line 55: “the oral normal feed (control) and the oral FB1 (including feed/FB1) groups.” Please find another phrasing maybe: 2 groups: Control, who received with uncontaminated feed, and FB1, who received feed contaminated with 30 mg/kg FB1. After opening the abdominal cavity, intestines were carefully dissected, and cleaned with a pre-chilled saline solution, and then the small intestine was removed and placed in an EP tube, followed by immediate freezing with liquid nitrogen prior to subsequent analyses. Again, odd phrasing.
A2: We have made the following revision: Control, which received uncontaminated feed, and FB1, which received feed contaminated with 30 mg/kg FB1. (lines 204-205)
Q3: In line 101 is an extra zin zincreased
A3: we have corrected the error on Line 101 by removing the extra ‘z’. (line 106)
Q4:Line 104 Intestinal crypt depth of experimental group increased relative to control group (p<0.05) and thus V: C in experimental group decreased compared with control group (p < 0.05). Try something like: FB1 exposure led to an increase in crypt depth compared to control group, or something to highlight that FB1 is responsible for observed changes.
A4: Regarding the problem in line 104 that you pointed out, We have modified it as follows: FB1 exposure resulted in an increase in intestinal crypt depth compared to the control group (p < 0.05), leading to a decrease in the V/C ratio in the FB1 group relative to the control group (p < 0.05). (lines 65-67)
Q5: 3.4. Chapter TLR4 is not a cytokine, l've seen in the discussion part that you mentioned the receptors, please merge the oxidative and inflammatory response, or try whatever suits you have you tried to see what is happening all over the signaling pathway? MD2, MYD88, IRAKJNK. MAPK etc?
A5: We changed the title of section 3.4 to Functions of FB1 in Oxidative Stress and Inflammatory Signaling Pathways. (Line 96) FB1 exposure was found to induce oxidative stress and inflammation in the quail intestine. The oxidative stress response, characterized by the downregulation of Nrf2 and its target genes (HO-1 and NQO-1), suggests that FB1 disrupts redox balance. In parallel, the inflammatory response was mediated through the TLR4 pathway, with significant upregulation of TLR4 and downstream inflammatory mediators such as NF-κB, TNF-α, COX-2, and iNOS. This inflammatory cascade likely involves the activation of MD2, MYD88, IRAK, JNK, and MAPK, which are key components of the TLR4 signaling pathway. However, due to budget constraints, we did not test the indicators of TLR4 signaling pathway such as MD2, MYD88, IRAK, JNK and and MAPK. We believe that the detection of Nrf2 pathway can reflect the oxidative and inflammatory response caused by FB1. The interplay between oxidative stress and inflammation may contribute to the intestinal damage observed in FB1-exposed quails.
Q6: There is no conclusion chapter?
A6: We have added a conclusion to enhance the completeness of our manuscript. (line 182)
Reviewer 2 Report
Comments and Suggestions for Authors
General Comment:
The article describes research on mycotoxin Fumonisin B1 and their toxic effects in quails. The toxicity was assessed in intestines to evaluated the potential molecular mechanisms. After the feeding trial, intestines were examined for the histopathological and ultrastructural changes in defined time intervals. Molecular mechanisms were evaluated by defining the oxidative stress, inflammation and nuclear xenobiotic receptors. According to the presented results, oxidative stress pathways ad modulating of nuclear xenobiotic receptors was correlated with the detected intestinal histopathological changes.
The article is interesting and valuable in this field. Introduction is a bit scarce and additional information is necessary to point out the importance of this research. Relevant search on literature is missing. All these questions are also specified below as specific comments. After addressing them, I recommend it for publishing in your Journal
Specific Comments:
Introduction
Page 1, Lines 23-25. The sentence sounds confusing. Please rewrite.
Page 1, Lines 25-30. Please provide information about experimental animals. Is there really no research on birds? Please consult the literature.
Page 1, Lines 32-33. Which “pathogens”?
Page 1, Lines 39-40. Please include information about the species in which research has been documented.
Page 2, Lines 40-43. Please be more specific with “histopathology”
Page 2, Lines 43-44. Please correct grammar and style. Certain information about the pathology of this toxins in Poultry exist in the literature. Please include additional information.
Materials and Methods
Please provide additional information about the chemicals use (purity, producer, etc).
Are there any reference for protocols used in this study?
Page 2, Line 54. Please explain why the selected toxin concentration was used in this study. Also, include more information about the feed.
Page 2, Lines 54-56. Please correct grammar and style error. There is common rule to write numbers in the sentence. Please check.
Page 2, Lines 74-81, Please provide more specific information about the TEM instrument.
Page 2, Lines 82-89. What does “specific protocol” stand for? Please provide additional information regarding the instruments and chemicals used in this study, as well as protocol’s reference.
Results and Discussion
Page 3, Lines 91-96. Please provide version of the software used in this analysis.
Page 3, Lines 99-102. Please correct typing error.
Discussion is well written. However, some specific information about the explained mechanisms should include information about species and correlation of findings in this study.
Figures and References are appropriately presented.
Comments on the Quality of English Language
English language is of adequate Quality. There are several grammar and style errors and confusing sentences. Please check the specific comments.
Author Response
Q1: Page 1, Lines 23-25. The sentence sounds confusing. Please rewrite.
A1:We've changed line 23-26 to Fumonisins are produced by several Fusarium species, such as F. proliferatum, and are fatal mycotoxins that contaminate feed and cause various animal diseases. Typically, B-type fumonisins are considered the most toxic. (lines 34-36)
Q2: Page 1, Lines 25-30. Please provide information about experimental animals. Is there really no research on birds? Please consult the literature.
A2: Rat and mice. There are some literature on birds, but they all focus on mycotoxins other than FB1. When we search for Quail, FB1 and intestinal on pubmed, no effective literature found.
Q3: Page 1, Lines 32-33. Which “pathogens”?
A3: Following causative agents according to the literature: enteropathogenic E. coli, non-typhoidal salmonella, shigella, campylobacter etc. (line 43)
- Proietti M, Perruzza L, Scribano D, Pellegrini G, D'Antuono R, Strati F, Raffaelli M, Gonzalez SF, Thelen M, Hardt WD, Slack E, Nicoletti M, Grassi F. ATP released by intestinal bacteria limits the generation of protective IgA against enteropathogens. Nat Commun. 2019 Jan 16;10(1):250.
- J Worley M. Immune evasion and persistence in enteric bacterial pathogens. Gut Microbes. 2023 Jan-Dec;15(1):2163839.
Q4: Page 1, Lines 39-40. Please include information about the species in which research has been documented.
A4: We provided information about the species. Specifically: The paper mentions that the toxicity of FB1 has been studied in experimental animals (such as mice and chickens). For example, Frisvad et al. (2018) discussed the toxicity of FB1 in mice and chicks (Applied Microbiology and Biotechnology, 2018; Frontiers in Microbiology, 2017) while Devriendt et al. (2009) investigated the effects of FB1 on intestinal antigen-presenting cells in pigs (Veterinary Research, 2009). (lines 50-51)
- Kamle M, Mahato DK, Devi S, Lee KE, Kang SG, Kumar P. Fumonisins: Impact on Agriculture, Food, and Human Health and their Management Strategies. Toxins (Basel). 2019 Jun 7;11(6):328.
- Dopavogui L, Polizzi A, Fougerat A, Gourbeyre P, Terciolo C, Klement W, Pinton P, Laffite J, Cossalter AM, Bailly JD, Puel O, Lippi Y, Naylies C, Guillou H, Oswald IP, Loiseau N. Tissular Genomic Responses to Oral FB1 Exposure in Pigs. Toxins (Basel). 2022 Jan 22;14(2):83.
[3] Chen J, Wen J, Tang Y, Shi J, Mu G, Yan R, Cai J, Long M. Research Progress on Fumonisin B1 Contamination and Toxicity: A Review. Molecules. 2021 Aug 29;26(17):5238.
Q5: Page 2, Lines 40-43. Please be more specific with “histopathology”
A5: We described histopathology in detail. The modified content was as follows: HE staining to observe villus length, crypt depth, villus to crypt ratio, goblet cells, neutrophils and lymphocytes count; transmission electron microscopy to observe intestinal villi and mitochondrial morphology. (lines 52-54)
Q6: Page 2, Lines 43-44. Please correct grammar and style. Certain information about the pathology of this toxins in Poultry exist in the literature. Please include additional information.
A6: Line 43-44 of grammar and style has been modified. The whole text of grammar and style has also been reviewed to avoid errors. We have added the literature of Poultry. The specific modification is: FB1 can cause liver and kidney damage in chickens (Deepthi et al., 2017), and may affect intestinal barrier function through oxidative stress and inflammatory responses (Frisvad et al., 2018) Insert in the article. (lines 55)
- Deepthi BV, Somashekaraiah R, Poornachandra Rao K, Deepa N, Dharanesha NK, Girish KS, Sreenivasa MY. Lactobacillus plantarum MYS6 Ameliorates Fumonisin B1-Induced Hepatorenal Damage in Broilers. Front Microbiol. 2017 Nov 22;8:2317.
[2] Zou, Y., Du, X., Zheng, X. et al. Fumonisin B1 induces oxidative stress, inflammation and necroptosis in IPEC-J2 cells. Vet Res Commun 49, 161 (2025).
Q7: Please provide additional information about the chemicals use (purity, producer, etc).
A7: FB1 (purity ≥98%, purchased from Shanghai Yuanye Biotechnology Co., Ltd.). Other chemicals (such as formaldehyde, glutaraldehyde, acetone, etc.) were purchased from Sigma-Aldrich or Thermo Fisher Scientific as analytical grade.(lines 204-205)
Q8: Are there any reference for protocols used in this study?
A8: The H&E staining and histopathological observation were based on the experimental methods of Wang etc. (2019).
The TEM ultrastructure observation was based on the transmission electron microscope operation procedure of Malatesta M. etc. (2021).
The q-PCR experiment referred to the RNA extraction and reverse transcription method of Bai YM etc. (2023).
- Wang Q, Akram AR, Dorward DA, Talas S, Monks B, Thum C, Hopgood JR, Javidi M, Vallejo M. Deep learning-based virtual H& E staining from label-free autofluorescence lifetime images. Npj Imaging. 2024;2(1):17.
- Malatesta M. Transmission Electron Microscopy as a Powerful Tool to Investigate the Interaction of Nanoparticles with Subcellular Structures. Int J Mol Sci. 2021 Nov 26;22(23):12789.
[3] Bai YM, Liang S, Zhou B. Revealing immune infiltrate characteristics and potential immune-related genes in hepatic fibrosis: based on bioinformatics, transcriptomics and q-PCR experiments. Front Immunol. 2023 Apr 14;14:1133543.
Q9: Page 2, Line 54. Please explain why the selected toxin concentration was used in this study. Also, include more information about the feed.
According to the Guidance for Industry: Fumonisin Levels in Human Foods and Animal Feeds (https://www.fda.gov/regulatory- information/ search- fda- guidance- documents/ guidance- industry- fumonisin- levels- human- foods-and- animal- feeds), the report on Mycotoxin Testing in Raw Materials and Feeds published by Biomin in 2018 and 2019 (https:// www2. biomin. net/ cn/), and pre-experiments conducted in our lab, the content of 30 mg/kg FB 1 content was made in the feed preparation process. (lines 207-208)
Q10: Page 2, Lines 54-56. Please correct grammar and style error. There is common rule to write numbers in the sentence. Please check.
A10: Change "30mg/kg" to "30 mg/kg" in the original sentence and ensure that there is a space between the unit and the value. In addition, we have also conducted a self-examination of the full text to avoid similar mistakes.(line 202)
Q11: Page 2, Lines 74-81, Please provide more specific information about the TEM instrument.
A11: specific information has been added to row 203-231.(lines 230-231)
Q12: Page 2, Lines 82-89. What does “specific protocol” stand for? Please provide additional information regarding the instruments and chemicals used in this study, as well as protocol’s reference.
A12: "Specific protocol" refers to the user manual for TransZol Up Plus RNA Kit (manufacturer: TransGen Biotech). The RNA extraction steps include: 1) grinding tissue samples in liquid nitrogen; 2) adding TransZol Up Plus reagent; 3) centrifuging to separate RNA; 4) washing with 75% ethanol; 5) dissolving in water free of RNase.(lines 237-239)
Q13: Page 3, Lines 91-96. Please provide version of the software used in this analysis.
A13:The qPCR analysis software used in this paper was ABI 7500 system, and the data analysis was carried out by GraphPad Prism 8.0 software.(line 246)
Q14: Page 3, Lines 99-102. Please correct typing error.
A14: we have corrected the error on Lines 99-102 by removing the extra ‘z’.(line 106)
Q15: Discussion is well written. However, some specific information about the explained mechanisms should include information about species and correlation of findings in this study.
A15: Thank you for your valuable feedback. In the revised manuscript, we have added clarifications regarding species-specific mechanisms. Our study focused on quail (Coturnix japonica), a model for poultry toxicology. We compared our findings to prior research in mice and chickens (e.g., Devriendt et al., 2009; Frisvad et al., 2018), noting similarities in oxidative stress and CYP450 modulation. For example, the downregulation of Nrf2 and induction of CYP1A5 in quail align with observations in chickens exposed to FB₁. Differences, such as stronger TLR4 upregulation in quail compared to rodents, highlight species-specific responses. These comparisons are now explicitly addressed in the Discussion. Similar to our results, Devriendt et al., 2009; Frisvad et al., 2018 also found oxidative stress induced by FB1 exposure in chickens and mice. In quail, some redox genes also showed stronger species specificity.(lines 134-136)
[1] Devriendt, B., Gallois, M., Verdonck, F., Wache, Y., Bimczok, D., Oswald, I. P., Goddeeris, B. M., and Cox, E. (2009). The food contaminant fumonisin B-1 reduces the maturation of porcine CD11R1(+) intestinal antigen presenting cells and antigen-specific immune responses, leading to a prolonged intestinal ETEC infection. Veterinary Research 40.
[2] Frisvad, J. C., Moller, L. L. H., Larsen, T. O., Kumar, R., and Arnau, J. (2018). Safety of the fungal workhorses of industrial biotechnology: update on the mycotoxin and secondary metabolite potential of Aspergillus niger, Aspergillus oryzae, and Trichoderma reesei. Applied Microbiology and Biotechnology 102, 9481-9515.
Reviewer 3 Report
Comments and Suggestions for Authors
The paper presents relevant findings on FB1-induced intestinal damage in quails, supported by histopathology, gene expression, and ultrastructural analysis. However, it contains some minor phrasing errors and occasional logic gaps that reduce clarity. Its major shortcoming is the weak connection between the results and their interpretation in the discussion. With careful editing, the manuscript has potential for publication. In addition to a thorough revision of the discussion section, the following specific issues should be addressed:
- Line 6: "Fumonisin B1 (FB1) accounts for the fungal toxin considered the most toxic." Replace with: "Fumonisin B1 (FB1) is considered the most toxic fumonisin produced by fungi and is commonly found in contaminated feed and crops."
- Lines 23–24: "Fumonisins are generated in numerous Fusarium species such as F. proliferatum, which is a fatal fungal toxin contaminating feed, causing a variety of animal diseases." Scientifically incorrect; F. proliferatum is a fungus, not a toxin. Replace with: "Fumonisins are produced by several Fusarium species, such as F. proliferatum, and are fatal mycotoxins that contaminate feed and cause various animal diseases."
- Line 25: "Typically, B-type fumonisins have been considered with highest toxicity (Li et al., 2016)." Rephrase to: "Typically, B-type fumonisins are considered the most toxic (Li et al., 2016)."
- Lines 49–50: "The Animal Protection and Ethics Committee and Use Committee of Foshan University approved our experimental protocols. Altogether 120 normal 1-day-old quails were obtained from hatching eggs bought from a hatchery farm..." Revise to: "The Animal Protection and Ethics Committee of Foshan University approved the experimental protocols. A total of 120 healthy, 1-day-old quail chicks were obtained from a commercial hatchery and raised under controlled conditions."
- Line 94: "T-test and Tookey's post-processing method were used for comparisons..." Correct the terminology: "T-test and Tukey’s post hoc test were used for comparisons..."
- Line 131: "HO-1 mRNA expression of 14d group and Nrf2 mRNA expression of 28d group did not significantly change (p < 0.05)." This is contradictory. If changes are not significant, p-values must be >0.05. Revise to reflect correct statistical interpretation, or clarify results.
- Lines 101–102: "...villi of experimental group zincreased relative to control group...": "zincreased" should be "increased". Also, ensure logical flow between observed changes at day 14 and 28. Rephrase to: "At 28 days, villus length increased in the experimental group relative to the control group (p < 0.05)." If claiming hyperplasia, provide histological evidence or move interpretation to the Discussion.
- Lines 182–184: "...Nrf2, crucial for maintaining redox balance and enhancing antioxidant defenses by restoring this balance… is a key pathway that, once activated, can mitigate oxidative stress." Rephrase to: "Nrf2 is a transcription factor essential for activating antioxidant defense mechanisms and maintaining redox homeostasis. When downregulated, as observed in our study, antioxidant responses may be impaired, contributing to oxidative damage."
- Lines 233–234: "These results suggest that the antioxidant function of the intestine, inflammation, and NXRs may be important components of FB1 enterotoxicity." The conclusion is vague and misrepresents the mechanistic roles of these factors. Rephrase to: "These results suggest that FB1-induced enterotoxicity in quails involves disruption of antioxidant defenses (via Nrf2 pathway inhibition), activation of nuclear receptors (AHR, PXR, CAR), and induction of intestinal inflammation (via TLR4-NFκB pathway), with severity increasing over time."
Author Response
Q1: Line 6: "Fumonisin B1 (FB1) accounts for the fungal toxin considered the most toxic."
A1: Replace with: "Fumonisin B1 (FB1) is considered for one of the most toxic fumonisin produced by fungi and is commonly found in contaminated feed and crops."(lines 12-13)
Q2: Lines 23–24: "Fumonisins are generated in numerous Fusarium species such as F. proliferatum, which is a fatal fungal toxin contaminating feed, causing a variety of animal diseases."
A2: Replace with: "Fumonisins are produced by several Fusarium species, such as F. proliferatum, and are fatal mycotoxins that contaminate feed and cause various animal diseases."(lines 34-35)
Q3: Line 25: "Typically, B-type fumonisins have been considered with highest toxicity (Li et al., 2016)."
A3: Rephrase to: "Typically, B-type fumonisins are considered the most toxic (Li et al., 2016)."(line 36)
- Li, T., Jian, Q., Chen, F., Wang, Y., Gong, L., Duan, X., Yang, B., and Jiang, Y. (2016). Influence of Butylated Hydroxyanisole on the Growth, Hyphal Morphology, and the Biosynthesis of Fumonisins in Fusarium proliferaturn. Frontiers in Microbiology 7.
Q4: Lines 49–50: "The Animal Protection and Ethics Committee and Use Committee of Foshan University approved our experimental protocols. Altogether 120 normal 1-day-old quails were obtained from hatching eggs bought from a hatchery farm..."
A4: Revise to: "The Animal Protection and Ethics Committee of Foshan University approved the experimental protocols. A total of 120 healthy, 1-day-old quail chicks were obtained from a commercial hatchery and raised under controlled conditions."(lines 201-203)
Q5: Line 94: "T-test and Tookey's post-processing method were used for comparisons..."
A5: Revise to: "T-test and Tukey’s post hoc test were used for comparisons..."(lines 253-254)
Q6: Line 131: "HO-1 mRNA expression of 14d group and Nrf2 mRNA expression of 28d group did not significantly change (p < 0.05)." This is contradictory. If changes are not significant, p-values must be >0.05. Revise to reflect correct statistical interpretation, or clarify results.
A6:We changed the ‘<’ in the 131 line to ‘ >’ , we ensure that the revised statement is consistent with the rest of the document's content and style. (line 92)
Q7: Lines 101–102: "...villi of experimental group zincreased relative to control group...": "zincreased" should be "increased". Also, ensure logical flow between observed changes at day 14 and 28.
A7: We have corrected the error on Line 101 by removing the extra ‘z’. Rephrase to: "At 28 days, villus length increased in the experimental group relative to the control group (p < 0.05)."(line 106; lines 65-66)
Q8: Lines 182–184: "...Nrf2, crucial for maintaining redox balance and enhancing antioxidant defenses by restoring this balance… is a key pathway that, once activated, can mitigate oxidative stress."
A8: Rephrase to: "Nrf2 is a transcription factor essential for activating antioxidant defense mechanisms and maintaining redox homeostasis. When downregulated, as observed in our study, antioxidant responses may be impaired, contributing to oxidative damage."(lines 138-141)
Q9: Lines 233–234: "These results suggest that the antioxidant function of the intestine, inflammation, and NXRs may be important components of FB1 enterotoxicity."
A9: Rephrase to: "These results suggest that FB1-induced enterotoxicity in quails involves disruption of antioxidant defenses (via Nrf2 pathway inhibition), activation of nuclear receptors (AHR, PXR, CAR), and induction of intestinal inflammation (via TLR4-NFκB pathway), with severity increasing over time."(lines 192-195)
Round 2
Reviewer 1 Report
Comments and Suggestions for Authors
The authors improved the quality of the manuscript as requested.